# In Silico Screening of Natural Flavonoids against 3-Chymotrypsin-like Protease of SARS-CoV-2 Using Machine Learning and Molecular Modeling

**DOI:** 10.3390/molecules28248034

**Published:** 2023-12-10

**Authors:** Lianjin Cai, Fengyang Han, Beihong Ji, Xibing He, Luxuan Wang, Taoyu Niu, Jingchen Zhai, Junmei Wang

**Affiliations:** School of Pharmacy, University of Pittsburgh, Pittsburgh, PA 15261, USA; lic154@pitt.edu (L.C.); feh34@pitt.edu (F.H.); bej22@pitt.edu (B.J.); xibing.he@pitt.edu (X.H.); luw49@pitt.edu (L.W.); tan77@pitt.edu (T.N.); jiz183@pitt.edu (J.Z.)

**Keywords:** 3-chyomotrypsin-like protease (3CL-pro), main protease (M-pro), SARS-CoV-2, long-COVID, flavonoids, molecular modeling, machine learning-based scoring function (ML-based SF), ligand–residue interaction profiles, structure-based virtual screening (SBVS), molecular dynamics simulation

## Abstract

The “Long-COVID syndrome” has posed significant challenges due to a lack of validated therapeutic options. We developed a novel multi-step virtual screening strategy to reliably identify inhibitors against 3-chymotrypsin-like protease of SARS-CoV-2 from abundant flavonoids, which represents a promising source of antiviral and immune-boosting nutrients. We identified 57 interacting residues as contributors to the protein-ligand binding pocket. Their energy interaction profiles constituted the input features for Machine Learning (ML) models. The consensus of 25 classifiers trained using various ML algorithms attained 93.9% accuracy and a 6.4% false-positive-rate. The consensus of 10 regression models for binding energy prediction also achieved a low root-mean-square error of 1.18 kcal/mol. We screened out 120 flavonoid hits first and retained 50 drug-like hits after predefined ADMET filtering to ensure bioavailability and safety profiles. Furthermore, molecular dynamics simulations prioritized nine bioactive flavonoids as promising anti-SARS-CoV-2 agents exhibiting both high structural stability (root-mean-square deviation < 5 Å for 218 ns) and low MM/PBSA binding free energy (<−6 kcal/mol). Among them, KB-2 (PubChem-CID, 14630497) and 9-*O*-Methylglyceofuran (PubChem-CID, 44257401) displayed excellent binding affinity and desirable pharmacokinetic capabilities. These compounds have great potential to serve as oral nutraceuticals with therapeutic and prophylactic properties as care strategies for patients with long-COVID syndrome.

## 1. Introduction

COVID-19 has been the most severe pandemic outbreak in the recent decade, with the global spreading transmission, high infection rate, and alarming death toll due to severe acute respiratory syndrome coronavirus-2 (SARS-CoV-2) [1,2]. This concerning plight was further worsened by the higher transmissibility and significant immune escape potential of the new strains [2,3]. Also, growing evidence suggests that many people develop chronic conditions that persist for an uncertain period after SARS-CoV-2 infections, known as post-COVID conditions (PCCs), or long COVID syndrome [4,5]. This high-risk COVID-19 group was termed as the “long-haulers”, who might experience various symptoms from light fatigue to severe neuropsychiatric symptoms [6,7]. Currently, long COVID syndrome has become the most worrisome challenge, with its mechanisms remaining unclear and insufficient treatment options available [8,9]. Therefore, in addition to the development of next-generation vaccines, the development of broad-spectrum antiviral agents against SARS-CoV-2 is a long-term viable therapeutic strategy.

The 3-chymotrypsin-like protease (3CL-pro), also known as main protease (M-pro), is a pivotal enzyme during viral life cycling found in coronavirus. With more cleavage sites than other papain-like proteases, 3CL-pro is essential for the generation of abundant important non-structural proteins required for viral replication [10,11,12]. Another advantage making 3CL-pro an attractive drug target is its relatively lower rate of conservative mutations compared to that of other potential targets including the crafty spike protein which has exhibited ten mutation sites in the Omicron variant and two in the Delta strain [13]. The recent resolution of the X-ray crystal structure of 3CL-pro and its substrate binding pocket has guaranteed accurate virtual screening, such as structure-based virtual screenings (SBVSs) [14,15,16] and high-throughput in vitro repurposing screening [10,17]. Consequently, multiple peptidomimetic inhibitors based on 3CL-pro were emerging and an oral agent called nirmatrelvir/ritonavir (PAXLOVID^TM^) developed by Pfizer has succeeded in the market after clinical trials [18]. Nevertheless, the ritonavir contained in Paxlovid is a potent inhibitor against the cytochrome P450 (CYP) 3A4 enzyme, making it prone to potential pharmacokinetic (PK) interactions with a range of drugs that are CYP3A4 substrates, inducers, and inhibitors [19,20]. Given that many of those drugs are widely prescribed to high-risk COVID-19 groups, this dilemma presents inevitable difficulties in clinical practice and implementation. For instance, a comprehensive evaluation of co-administered drug usages, along with individual patient factors should be conducted before prescribing Paxlovid. Meanwhile, special considerations like contra-indication, careful monitoring, and dose adjustment are warranted for over 120 drugs to avoid adverse drug reactions, according to the guidelines of regulatory agencies [20,21].

The use of immune-boosting foods and nutrients represents a safe alternative to standard therapies with preventative and therapeutic interventions for COVID-19 and PCCs in high-risk populations [8]. Flavonoids constitute a large scale of food nutrients (e.g., tea and citrus fruit) and plant metabolites, presenting multifunctional and broad-spectrum antiviral effects against COVID-19 over existing synthetic drugs [22,23,24]. Several flavonoids exhibited inhibitory activity against SARS-CoV-2 by binding to essential viral targets required for virus entry and/or replication [22,25]. Also, flavonoids have demonstrated significant immunomodulatory activities to alleviate excessive immune responses and, thereby, ameliorate long-COVID-19 syndrome [22]. Moreover, with desirable safety profiles, flavonoids demonstrated a potential for use as promising nutraceutical products in and out of pregnancy to mitigate the risk of fetal brain damage during maternal COVID-19 [23,26]. Unlike other pharmacological treatments, they can be consumed through a dietary approach with better tolerance, without the need for specific manufacturing processes [27]. Despite many advantages, the diversity of multidimensional chemical structures has made it difficult to identify promising flavonoid candidates from various plausible scaffolds. 

Considering the experimental assay screening process is time-consuming and laborious, virtual screening with molecular-binding descriptors accelerates the “hits” screening process, which is a much preferable tactic, especially in public health crises [28]. The classical scoring function (SF) with predetermined function forms must reach a compromise between computation cost and accurate prediction performance [29,30]. It also hardly characterizes the complicated heterogeneity of the drug-target interactions (DTI) due to simplifications in protein-ligand recognition for the sake of efficiency [31,32]. Recently, the nonparametric Machine Learning (ML) approaches have been widely applied in drug discovery and development [33,34]. The ML-based SF turned out to be an innovative way to improve the screening performance of SBVSs, utilizing a variety of descriptors including geometrical features, energy terms, and pharmacophore features [35,36,37]. Our research team developed the interaction profiles (IPs), novel molecular descriptor that incorporates the calculated ligand–residue interaction energies. Encouragingly, this IP-based scoring function (IP-SF) trained using various ML algorithms significantly outperforms the traditional Glide SF in terms of scoring, ranking, and screening power when tested ligands against six drug targets [32]. Thus, the virtual screening approach using ML-based IP-SF models is appropriate for the identification of flavonoids from diverse chemical structures. In addition to SBVS, molecular docking and dynamic simulations studies can determine the interaction of a specific molecule with a certain protein successfully in a period. Furthermore, for orally administered drugs and nutraceuticals, properties like absorption, distribution, metabolism, excretion, and toxicity (ADMET) profoundly affect clinical efficacy and safety. This makes ADMET assessment a necessary step in identifying druglike flavonoids with in-silico screenings.

Herein, we conducted molecular modeling to obtain IP-SFs which have been used as input features for ML algorithms for three different databases, including two DTI databases for 3CL-pro binding assays and one flavonoid metabolites database. Figure 1 describes this virtual screening workflow. First, three different compounds datasets in Table 1 were prepared after data collection, processing, and qualification. Next, both regression and classification ML models were constructed utilizing two different training datasets. We experimented with an ensemble approach to obtain consensus-based model prediction by combining outputs from separate ML models using majority voting to obtain consensus. Consequently, building on the ensembled regressor and classifier, two consensus-based ML predictions (i.e., binding energy and biofunction) on flavonoids were made to obtain the preliminary screening of inhibitor hits. ADMET analysis was then conducted to assess the drug-likeness properties of the filtered hits and ensure bioavailable potential. Lastly, a molecular dynamics (MD) simulation was implemented to investigate the dynamic behavior of the prominent flavonoids hits and confirm their appropriate bindings in the aspects of stability and affinity. The long-term objective is to develop the potential pharmacological candidates and/or oral nutraceuticals to alleviate long COVID-19 syndrome as a supportive adjunctive intervention for the current antiviral treatment strategy.

## 2. Results and Discussion

### 2.1. Dataset Preparation

After the same procedures of data processing and qualification for collected compounds (the first step in Figure 1), the IPs were generated for three datasets for different usages. The numbers of training set compounds after cleanup are 1011 for Training Set A and 6059 compounds for Training Set B (Table 1). In total, 57 interacting residues out of a total of 306 residues in each 3CL-pro monomer (PDB: 6M2N) were identified based on the pre-defined threshold to participate in modeling with ML, serving as the input feature representations (i.e., molecular descriptors). Appendix A lists the correspondence between the descriptor ID used for machine learning and the original residue IDs in PDB for 3CL-pro. Appendix A provide the interaction energies profiles (kcal/mol) between the residues and compounds from three different datasets.

For both training sets, the amount of active and inactive data is dramatically skewed, as shown in Table 1 (N_decoys_/N_actives_ ≈ 10 for Training Set A, N_decoys_/N_actives_ ≈ 20 for Training Set B). With an imbalanced dataset, models will be biased towards predicting the majority class, leading to poor performance in the minority class, even with high overall accuracy [40,41]. Also, Weiss and Provost found that under-sampling the over-represented classes was an effective technique for handling class imbalance and reduced the risk of overfitting the abundant majority class [41]. Hence, a “rule-based under-sampling” step was implemented to ensure a balanced distribution of actives and decoys [42]. Consequently, 10 subsets (A1–A10) of the Training Set A and 20 subsets (B1–B20) of the Training Set B were generated for the regression and classification models training, respectively, with the under-represented actives matching the inactives in each training subset. 

### 2.2. ML-Based Model Performance

Three training trials were repeated for each subset, and, hence, in total 30 trained regression models were generated from training subsets A1–A10, and 60 trained classification models were generated from subsets B1–B20. For binding free energy prediction, the 10 models with the lowest predicted RMSE (Reg1–Reg10) were selected to ensemble the regressor. While for bioactivity score prediction, the 20 classification models with the highest accuracy (Cla1–Cla20) were directly selected. Additionally, out of the remaining forty classification models, five additional classifiers (Cla21–Cla25) with the lowest false positive rates (FPR < 10%) were also added to construct the final “ensembled classifier” (Figure 1). The number of compounds, ML algorithms, and all metrics values for the cross-validation results for each regression model and classifier are presented in Appendix A. 

Figure 2 shows that half of the regression models adopted the algorithm of Exponential Gaussian Process Regression (GPR), suggesting it outperforms other algorithms in terms of RMSE in five different subsets (Appendix A). Meanwhile, Bagged Trees (BT) was the algorithms frequently applied in 25 classifiers, followed by K-Nearest Neighbors (KNN), which achieved the best accuracy among subsets B1–B20. The pairs of specific classifier algorithms and subsets are presented in Appendix A. The cross-validation results in Figure 3 and Figure 4A (blue bars) indicated that all models attained reasonable predictive performance, with greater than 1.5 kcal/mol RMSE for regression models and above 60% accuracy for classifiers. As shown in Figure 4B, it is noticeable that the Support Vector Machine (SVM) classifiers (Cla21–Cla25) displayed a markedly lower false positive rate compared to other classifiers. The incorporation of these five additional models greatly improves the ability to detect the false positive hits against 3CL-pro and thereby elevated comprehensive accuracy when validated in the complete Training Set B (>90% ACC, orange bars in Figure 4A).

Regarding the performance of the ensembled regressor, the consensus-based validation results in the complete Training Set A displayed a low RMSE of 1.18 kcal/mol for the binding energy prediction (Figure 3), using a set of ML algorithms including GPR, SVM, and BT. Meanwhile, the validation results in Figure 4 demonstrated that the consensus of multiple ML classifiers (SVM, KNN, BT) achieved a high screening power with an accuracy of 93.9% and a false positive rate of 6.4%. Since the number of active compounds is only 5.7% of the inactive ones in the original complete Training Set B (Table 1), we only focused on the errors for inactive predictions for model performance. Thus, a lower FPR is required as it suggests that the model has minimized the error of labelling the inactive compound as the hits and the overall screening power has been augmented. The trade-off of a high false negative rate is acceptable, by contrast, because our Prediction Set has large amounts of flavonoids and ML-based virtual screening served as a preliminary screening test for any feasible hits at all costs. Moreover, we prepared another screening test based on regression models to capture or save those potential active compounds that were labeled as decoys falsely. Therefore, our multi-model consensus prediction can significantly improve the success rate of identifying true antiviral compounds (hits) targeting 3CL-pro.

### 2.3. Flavonoid Hits Screening

A flavonoid is recognized as an inhibitor only when it can satisfy one of the following three conditions based on consensus-based ML predictions: (1) its predicted binding energy is equal or less than −7.4 kcal/mol; (2) its predicted bioactivity score is equal or greater than 0.8, i.e., at least 20 out of 25 classifiers voted “1”; (3) it belongs to the top 10% flavonoids ranked by both binding energy and bioactivity score. Binding energies are ranked from lowest to highest, while bioactivity scores are ranked from largest to smallest. Through ML consensus-based virtual screening, 120 flavonoids were rapidly screened out and identified as the potential hits target at 3CL-pro. The original prediction results of regression models and classifiers for all 6001 flavonoids are provided in Appendix A.

The specific risk scores and codes generated using the ADMET Predictor for all 120 screened hits are provided in Appendix A. According to the strict “dumb” ADMET filtering, only 50 of the drug-like hits were retained without severe ADMET risks. Boxplots in Figure 5 compare the risk scores distribution of seven risk models for the flavonoid hits before and after the filtering steps being carried out. The full consensus-based ML predicted results and ADMET risk information for the top120 and top50 hits were integrated and are summarized in Appendix A. The drug-likeness concept can qualitatively determine the oral bioavailability and, PK profile, as well as the potentially toxic and mutagenic risk of a compound, based on their molecular properties. Poor PK and ADMET properties accounted for up to 50% of attrition in drug development in the 1990s [43,44,45]. However, the implementation of early ADMET screening with more accurate modeling and prediction approaches has significantly reduced these failures, with around 10% dropouts due to PK reasons in the 2000s [45]. Hence, assessment of the ADMET and PK profile early in drug discovery can guide hits selection and optimization to avoid downstream failures. 

### 2.4. Molecular Dynamics Analysis

Three types of Root-Mean-Square-Deviation (RMSD) were calculated during the binding simulation of these 50 flavonoids: the 3CL-pro protein, the ligand with the least square fit (LS Fit) and the ligand without the LS fit (No Fit). The ligand No Fit RMSD represents the overall deviation (including translation and rotation) of ligands to their initial conformations, while the LS Fit process aligned ligands to the initial ligand conformations reducing the influence of ligand translation and rotation during MD. Therefore, the No Fit RMSD result shows the overall stability of ligands within the binding pocket, and the LS Fit RMSD results only measure the ligand stability of intramolecular conformational changes. 

By setting the cutoff of the maximum value of the No Fit RMSD at 5 Å, 18 molecules passed the filter with good binding affinities. We then applied an additional PBSA binding free energy threshold, −6.0 kcal/mol, to select the top9 flavonoids as listed in Table 2. Among them, KB-2 (**PubChem CID: 14630497**) displayed the most stable ligand binding and was picked up as a top hit, having both the lowest values of MM/PBSA energy (−9.89 kcal/mol) and the maximum RMSD (2.83 Å). In Figure 6, the black curve suggests the 3CL-pro protein undergoes little conformational changes with a mean RMSD value of around 1 Å. The RMSD distribution curves for “LS Fit” and “No Fit” demonstrates that most conformations had low RMSD values for both scenarios and the highest RMSD values remained below 3 Å. Thus, KB-2 formed favorable protein–ligand binding which was very stable during our MD simulation. The RMSD distribution graphs of the other eight top flavonoids are shown in Appendix A and the maximum RMSD values are listed in Table 2. The calculated PBSA-energy and RMSE values for all top50 druglike flavonoids are presented in Appendix A.

For molecular interaction pattern, hydrogen bonds (H-bonds) are the most common type of directed intermolecular force in biological complexes. They played an important role in the process of determining the specificity of molecular recognition. Figure 7 shows the H-bond interactions as well as the π–π stacking interactions within the representative KB-2 ligand–protein complex sampled by MD simulations. Specifically, KB-2 forms H-bond interactions with the residues THR26, HIS41, CYS44, and GLY143 (violet arrow), which could be crucial for its binding to the 3CL-pro. Additionally, Pi–Pi contact was also observed at the site of HIS41 residue (green arrow).

### 2.5. Top Flavonoids

We identified nine prioritized bioactive flavonoids (Figure 8) from plants as promising anti-SARS-CoV-2 agents hits through multiple in-silico tools, including ML-based IP-SF prediction, drug-like properties, ADMET filtering, and structural stability and binding free energy analysis. They belong to six different classes of flavonoid structure (Table 3), including three flavanones, two isoflavonoids, one chalcone, one dihydroflavonol, one flavone, and one flavonol. Regarding the drug-like properties prediction as an oral nutraceutical agent, all nine hits satisfied Lipinski’s rule of five. Additionally, there is no serious CYP metabolism risk indicating the drug-drug interaction issues could be circumvented. Among these newly identified hits, 9-O-Methylglyceofuran (**PubChem CID: 44257401**) and Oxyayanin A (**PubChem CID: 5281676**) have better drug-like properties with full ADMET risk scores lower than 3.0 (Table 2) and are likely to present an excellent PK profile. However, despite having the most stable and strongest binding affinity against the 3CL-pro catalytic site, KB-2 might have minor oral absorption issues (Appendix A). More efforts like molecular modification and optimization based on KB-2 structure are warranted. 

The top hits were screened from phytochemical flavonoid derivates, however, the extensive information on the specific protein targets is missing for most flavonoids. To address this issue, we employed TargetHunter, an online tool to search potential targets of small molecules by matching the query structure with reported bioactive compound–target pairs [46]. By setting a 2D similarity threshold of 80% and employing FP2 fingerprints, we were able to pinpoint several prospective targets of top flavonoids compounds. Notably, KB-2 (**PubChem CID: 14630497**) had an similarity score of 0.84 to CHEMBL1719948, a potent inhibitor of butyrylcholinesterase with an IC50 of 1.7 μM; Sophoraflavanone C’s (**Pub-Chem CID: 85403243**) showed a similarity score of 0.85 to CHEMBL1096939, another inhibitor of butyrylcholinesterase with comparable binding activity [47]; 3-*O*-demethyl-8′-Hydroxyrotenone (**PubChem CID: 44257401**) showed a similarity score of 0.86 to CHEMBL429023 which is a strong inhibitor against NADH-ubiquinone oxidoreductase chain 4 with an IC_50_ of 3.5 nM [47]; Uvaretin’s (**PubChem CID: 73447**) achieved a structural similarity score of 0.90 to CHEMBL254648 which was identified as a thrombin inhibitor with an IC_50_ of 12.3 μM [48]; and lastly Ovalichromene B (**PubChem CID: 10981007**) had a similarity score of 0.90 to CHEMBL147199, another thrombin inhibitor with an IC_50_ of 1.38 μM [49].

Kumari and Subbarao employed the deep-learning based virtual screening against 3CL-pro and they found nine out of ten prioritized active anti-SARS-CoV phytochemical compounds belonging to the flavonoids [50]. This finding galvanized our efforts to delve into which flavonoids exhibit superior anti-SARS-CoV2 activities. Figure 9 shows the four important screening criteria of flavonoid hits to quantitatily describe the binding ability agaisnts 3CL-pro. There is no obvious association between MM/PBSA calculated energy (Figure 9A) and ML-predicted binding energy (Figure 9C), considering the interaction energies were constructed from two diverse systems and approaches. Interestingly, Figure 9D shows that seven out of the nine top hits were predicted with a high bioactivity score (0.8) and ranked as the top 10% compounds based on the ML classifiers. In comparison with Figure 9C, they did not obey the screening criteria by regression models (the 2nd condition of ML tests). Nevertheless, the additional test of regression models indeed identified some molecules (e.g., Taxifolin 3-methyl ether and Dihydrotricetin) that were missed by the classifier tests due to low predicted bioactivity scores. This revealed the advantage of building both types of ML models simultaneously to perform virtual screening. Considering the two training datasets are mutually independent, they complemented strengths to building an IP-SF, and thus improved the power of screening potential 3CL-pro inhibitors, which was verified by the MD simulation results. Compared to the previous work on the ML-based SBVS of flavonoids [16,50], our study adopted a more comprehensive methodology, for instance, the ML models were constructed using much larger training data; multi-models were ensembled to attain consensus; both the molecular docking and MD simulations were applied. More importantly, our novel molecular descriptor, IP-SF which was derived from MM/PBSA free energy decomposition, can better characterize the heterogeneity of the ligand-target interactions, thus, allowing more accurate prediction of protein-ligand binding affinity [32].

Collectively, we conclude that KB-2 and 9-*O*-Methylglyceofuran exhibited both de-sirable binding affinity and pharmacokinetics capabilities. Our work would also contrib-ute to the rational drug design of flavonoid therapeutic agents according to the original phyto-chemical structures of the selected compounds from our computational studies. To the best of our knowledge, the relevance to COVID-19 of these top flavonoids from plants has not been reported before. To further validate the SARS-CoV-2 3CL-pro inhibitory activity and feasible druglike properties, in vitro enzymatic inhibitory assays and in vivo PK investiga-tions are warranted. In the future, we plan to conduct a more exhaustive exploration of the structural similarities between our identified flavonoids and other bioactives with known mechanisms of functions. This investigation could provide useful information on the as-sessment of those flavonoids as potential anti-SARS-CoV-2 agents in drug development.

## 3. Methods

### 3.1. Data Sources

The crystal structure of SARS-CoV-2 3CL protease was retrieved from the protein data bank, (PDB, www.rcsb.org (accessed on 12 January 2023)) with the entry ID of 6M2N. From the Chambly bioassay database, (ChEMBL, https://www.ebi.ac.uk/chembl/ (accessed on 12 January 2023)), a total of 1240 ligand compounds of 3CL-pro were retrieved and served as Training Set A for regression model training and construction [38]. As shown in Table 1, The continuous output was the binding energy in kcal/mol (ΔE_exp_), calculated from experimental half maximal inhibitory concentration (IC_50_) values, using Equation (1). However, only 101 actives had validly measured IC_50_ values, with calculated ΔE_exp_ all below −7 kcal/mol. The remaining 1010 inactive molecules were assigned binding free energies of −5.44 kcal/mol which corresponds to an IC_50_ value of 100 μM.
(1)$$∆E_{\mathrm{expt}}=-\mathrm{RTln}\left({\mathrm{IC}}_{50}\right)$$

In Training Set B, all 8702 compounds were involved in the SARS-CoV-2 3CL-pro enzymatic assay for primary screening [17]. The measured inhibitory activity type in primary screening is the normalized inhibition percent (**%** inhibition) to the positive control. Utilizing a cutoff of 25% of the precent inhibition compared to the control, we manually separated all compounds into the active set and inactive set (decoy set). This allowed the top 5% ligands to be labeled “active” and the rest “inactive”, constituting the categorical outputs for binary classification ML training. This less stringent cutoff setting will help to capture more potential actives, since the primary screening was performed at a single high compound concentration (20 μM), in which lower affinity but still active compounds may not reach saturation [17].

An open flavonoid metabolites database (http://metabolomics.jp/wiki/Category:FL, (accessed on 12 January 2023)) provides a collection of 6961 registered phytochemical structures in various plant species [39]. This database is a comprehensive resource with detailed information about identified flavonoid structures, plant species, and references to support research and analysis like virtual screening in the field of flavonoid chemistry and biology. It serves as the Prediction Set used for the IP-SF prediction and virtual screening of the potential inhibitors against 3CL-pro. 

### 3.2. Molecular Modeling Study and Interaction-Profile (IP) Calculation

#### 3.2.1. Molecular Docking and System Setup

We adopted the best computational protocol developed by Ji et al. for ligand-residue IPs construction, MIN + GB (applying minimization to relax the complex and applying a GB model to account for the solvent effect) [32]. It was demonstrated to have minimal calculation cost and better performance compared to other simulation protocols as well as the conventional docking algorithms in SBVS. The details of molecular docking and system setup can be found in the original paper [32]. 

To analyze the binding interactions between the ligand and individual residues, we performed an energy decomposition analysis based on the molecular mechanics generalized Born surface area (MM-GBSA) method [32]. The optimized ligand-receptor complex obtained was used as input for the MM-GBSA calculations. Binding free energies were computed between the ligand and each receptor residue using an in-house script that processes the energy components from the output of the Sander module of AMBER 22 software [51]. The dielectric constant was set to 1.0 for the interior protein environment and 80.0 for the exterior solvent environment. If a residue has a valid value of interaction energy (<−0.001 kcal/mol) with any ligand in three compound datasets, it is recognized as an interacting residue for characterizing the binding profile. The MM-GBSA interaction energies between all interacting residues and a ligand constitute the interaction profile (IP) of this ligand.

#### 3.2.2. Data Processing and Qualification

Notably, a small number of compounds were satisfied inevitably in the gaining IP-SF process, as presented in Table 1, because of failures or instabilities under the following structure processing protocols. Docking scores at the dimerization site were also calculated by the Ligand Docking module from the Glide program for all compounds from the two training sets. Compounds with more favorable docking scores at the dimerization site than at the catalytic site were omitted. This data qualification step in our workflow (Figure 1) enabled the identification of bona fide inhibitors that can not only target 3CL-pro but also inhibit its catalytic function. 

### 3.3. Machine Learning Model Construction and Prediction

#### 3.3.1. Model Training and Evaluation

The feature matrix consisted of MM-GBSA IPs between all ligands and 57 interacting residues, serving as the input descriptors for each ML algorithm. All 57 features are assumed to be equivalently important and no extra processing like principal component analysis was conducted before training since all input data were extracted following the same protocol and were at the same scale. The inputs, i.e., feature matrixes or interaction profiles, for training ML regression (A1–A10) and classification models (B1–B20), are provided in the Appendix A. All available ML model types with default parameters in the Regression Learner App and Classification Learner App, MATLAB (version R2022b, MathWorks Inc., Portola Valley, CA, USA), were trained parallelly with 20-fold cross-validation. In Classification Learner App, the misclassification costs have been customized so that the penalty score of false positives were doubled in comparison with false negatives. The performance metrics for regression models include root mean square error (RMSE), mean absolute error (MAE), and squared correlation coefficient (CORR2). The performance metrics for classifier evaluation comprise the area under curves (AUCs) of receiver operating characteristics (ROCs), accuracy (ACCs), and false positive rate (FPRs).

#### 3.3.2. Consensus-Based Model Prediction

The cross-validation results of each ML model provided the predictive performance in a corresponding balanced subset, which may not reflect the class imbalance that future predictions would encounter. To ensure a more robust screening, we adopted an ensemble technique by combining a variety of Machine Learning algorithms trained for different subsets to generate the decision-fusion of multiple models [42,52]. 

The performance metrics of these two consensus models, i.e., ensembled regressor and ensembled classifier, were evaluated using the complete Training Set A and complete Training Set B, accordingly. Specifically, all 10 regression models were applied to the complete Training Set A successively and a single predicted binding energy (ΔE_pred_) was generated based on the averaged result from 10 ΔE_pred_ outputs for each compound. Likewise, all 25 classifiers were utilized to yield a single predicted bioactivity score for each compound in Training Set B. We utilized the consensus score cutoff of 0.5 to determine a compound’s class, i.e., it belongs to the “0” class if the consensus score is smaller than 0.5 and the “1” class otherwise. Note that a consensus score of 0.5 is impossible as all the 25 classifiers predict either “0” or “1” and they contribute equally to the consensus score.

As shown in Figure 1, the constructed consensus models were then applied to the Prediction Set to obtain the consensus-based model predictions, comprising the mean value of the binding energy from 10 regression models, and the bioactivity scores from 25 classifiers.

### 3.4. ADMET Risk Filtering

ADMET Predictor^®^ (version 10.4, Simulation Plus, Inc., Lancaster, CA, USA) allows for a summary of potential pharmacokinetics and toxicity issues of a drug hit, by comparing various predictions to a series of risk rules and tallying violations. It presents a weighted ADMET risk score reflecting the number and severity of violations. The score thresholds for compound filtering were obtained by focusing on a specific subset of drugs in the World Drug Index the (WDI), removing the irrelevant class of compounds. For instance, the threshold of concern for full AMDET Risk is 7, as only 10% of a reference set of 2260 commercial drugs from WDI exceeded this value. As shown in Table 3, the seven ADMET Risk models were applied as strict “dumb” filters in this study, discarding any screened “hits” that violated any criteria exceeding any risk score threshold. The canonical SMILES of the selected molecular structure of the flavonoid hits were prepared as the model input.

### 3.5. Molecular Dynamics (MD) Simulation

Energy minimization was carried out for the 6M2N protein using Protein Prep Wizard with a default constraint of 0.3 Å of Root-Mean-Square-Deviation (RMSD) and the OPLS3 force field. The grids were generated by the Receptor Grid Generation package (http://gohom.win/ManualHom/Schrodinger/Schrodinger_2015-2_docs/maestro/help_Maestro/glide/receptor_grid_generation.html accessed on 6 November 2023) using the centroid of the initial ligand, baicalein, in 6M2N by defining the inner box to be a 10 Å cube, with the outer cube box lengths being 30 Å. Site-specific molecular docking for 50 flavonoid hits against protease was performed using the Ligand Docking module with default parameters and standard precision (SP) from the Glide program in Maestro (version 11.2, Schrödinger, Inc., New York, NY, USA) [53,54,55].

We collected the best ligand-receptor docking poses from docking processes and used them as the initial conformations for the subsequent molecular dynamics (MD) simulations to evaluate the dynamic properties and binding affinities. The simulation box consisted of one copy of the protein–ligand complex, 0.15 M NaCl, and ~16,000 explicit TIP3P water molecules [56]. The AMBER ff14SB forcefield from AMBER 22 was applied to describe the protein [51]. The ligands were described using the general AMBER force field (GAFF) and the Antechamber module in AMBER Tools was applied to generate the residue topologies and additional force field parameters [57]. The RESP par-tial charges were assigned to ligand atoms [58]. 

The MD simulations were conducted using the AMBER 22 program [51]. The systems were first relaxed through a set of minimization, heating, and equilibration cycles. The all-atom MD simulations were run for these 50 ligand-protein complex systems at the Center for Research Computing (CRC) at the University of Pittsburgh. Each ligand-protein system was subjected to a 225 ns production run. The stability of the receptor-ligand complex was evaluated by a calculation of the Root-Mean-Square-Deviation (RMSD) of the C-α relative to the initial structures. 

Snapshots for trajectories without solvents were extracted from the MD simulation to conduct MM-PBSA energy calculations, using both sander and pmemd programs for parameter preparation and Delphi V4 Release 1.1 for computing the polar part of the solvation free energy, as detailed below [59,60].

### 3.6. MM-PBSA Energy Calculation

The binding free energies of the protein–ligand complexes were computed based on Equation (2). The $\Delta E_{MM}$ term denotes the changes in molecular mechanics energy in the gas phase, $\Delta G_{sol}$ denotes the alteration in solvation free energy, and the TΔS corresponds to the entropy change of the ligand–receptor system during the ligand-binding process [61]:(2)$$\Delta G_{bind}=\Delta E_{MM}+\Delta G_{sol}-T\Delta S$$

Subsequently, polar and non-polar terms are used to further split the $\Delta G_{sol}$ part (Equation (3)). The Poisson–Boltzmann (PB) or Generalized Born (GB) models are commonly applied for computing the polar part, referred to as $\Delta G_{PB/GB}$. The nonpolar part $\Delta G_{SA}$ is calculated with solvent-accessible surface areas (SASAs) [62]. The contribution of conformational entropy during the binding process, *T*Δ*S*, was estimated using the WSAS program [48].
(3)$$\Delta G_{sol}=\Delta G_{PB/GB}+\Delta G_{SA}$$

## 4. Conclusions

In the current situation, it is still critical to discover compounds that can inhibit the viral function of SARS-CoV-2, as the potential dietary supplements mediating the long-COVID-syndrome. A novel multi-step virtual screening strategy was developed to identify bioavailable flavonoid nutrients that can target 3CL-pro which played a vital role in previous pandemic fighting. We utilized a ligand–residue interaction profile which adequately accounts for the heterogeneity of protein-ligand binding to construct ML models. We found that the multi-model consensus (the fusion of all model predictions) using various ML algorithms can significantly boost the prediction performance for both types of modeling by taking advantage of the complementary strengths of disparate models. Hence, 120 flavonoids were recognized as potential 3CL-pro inhibitors as they passed the ML tests consisting of 10 ML-trained regression models and 25 classifiers. Furthermore, the top hits were screened out for future research that displayed stable binding affinity during the 218-ns molecular dynamics simulations and had no serious AMDET risk. Overall, nine prominent flavonoids were identified with great potential to inhibit the 3CL-pro enzyme of SARS-CoV-2 and play a role in COVID-19 prevention and/or intervention. In the future, according to our recommendation, further in-vitro and in-vivo hits confirmation investigations of these compounds need to be carried out to validate their feasibility.

## Figures and Tables

**Figure 1 molecules-28-08034-f001:**
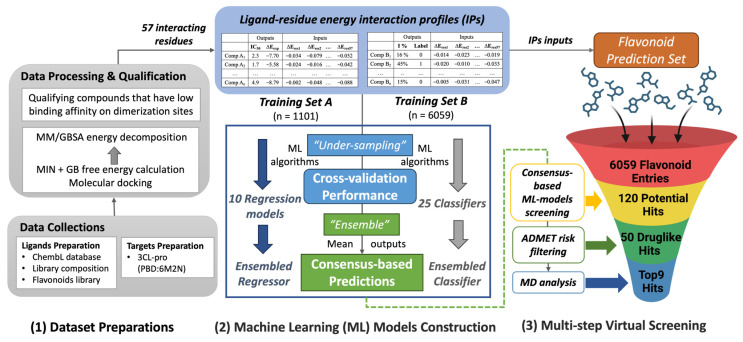
The in silico screening workflow developed in this study, consisting of three key steps: (**1**) dataset preparations; (**2**) Machine Learning models construction, and (**3**) multi-step virtual screening. Note that the ML models were trained, evaluated and selected for multi- subsets that were gen-erated using the “rule-based under-sampling” technique. Then the top-performed regression models were applied to predict a compound’s binding free energy, while the top-performed clas-sifiers were applied to predict the biofunction (active or inactive) of a compound. The ensem-bled-based prediction is expected to increase the accuracy and robustness of the prediction. The acronyms used in this figure are listed below: MD—molecular dynamics; MM/GBSA—molecular mechanics energies combined with the generalized Born (GB) and surface area continuum solvation; MIN—minimization; ADMET—absorption, distribution, metabolism, elimination and toxicity. ADME describes the drug-likeness properties for oral bioavailable agents without toxicity risk.

**Figure 2 molecules-28-08034-f002:**
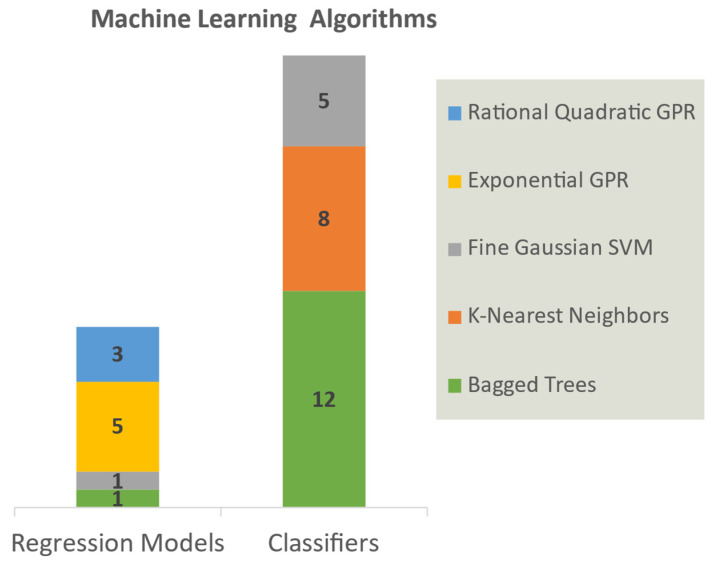
The Machine Learning (ML) algorithms for 10 regression models and 25 classifiers, presented in two stacked bar charts, respectively. Different colors represent different ML algorithms. The number on each color block represents the number of regression models or classifiers using this ML algorithm, with the details listed in Appendix A GPR, Gaussian Process Regression. SVM, Support Vector Machine.

**Figure 3 molecules-28-08034-f003:**
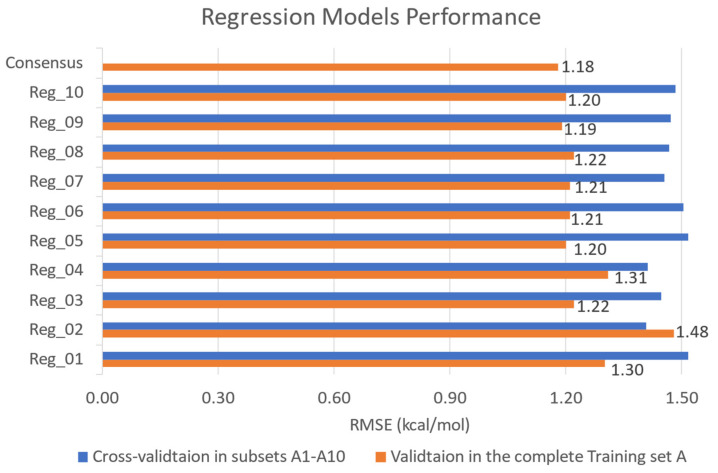
The predictive performance of 10 selected regression models (Reg1–10) and results of model consensus, presented in bar charts of RMSE for predicted binding energy (ΔE_pred_). The blue bars represent the cross-validation results of each model from its training subset and the values are provided in Appendix A. The orange bars represent the validation results tested in the original imbalanced Training Set A and the number above each original bar represents the RMSE value (kcal/mol).

**Figure 4 molecules-28-08034-f004:**
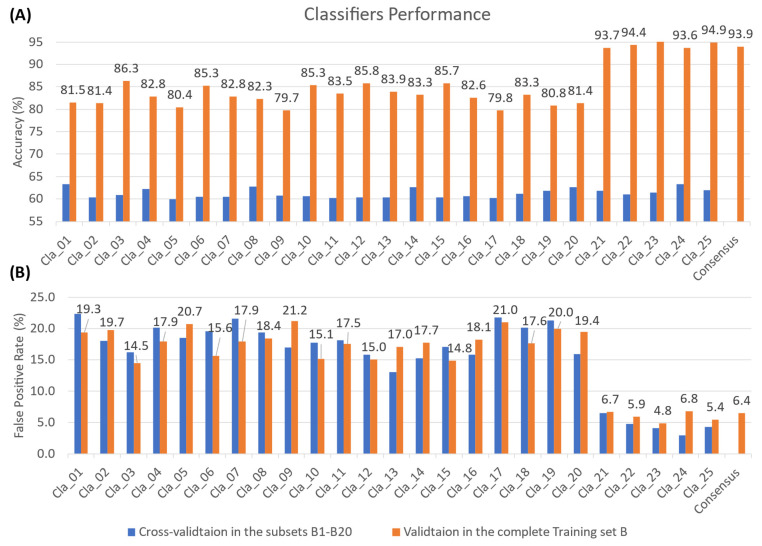
The predictive performance of 25 selected classifiers (Cla1–25) and consensus-based results, presented in bar charts of accuracy (**A**) and false positive rate (**B**) for predicted active labels. The blue bars represent the cross-validation results of each model from its training subset and the values are provided in Appendix A. The orange bars represent the validation results tested in the original complete Training Set B and the number above each orange bar represents the accuracy and false positive rate values (%).

**Figure 5 molecules-28-08034-f005:**
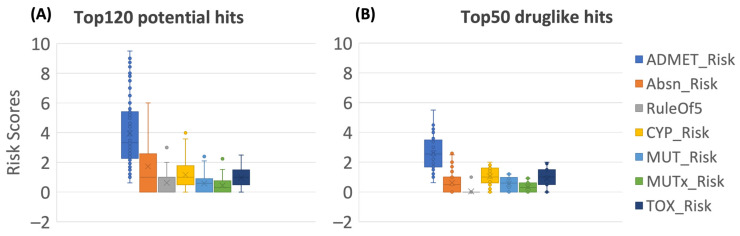
ADMET risk analysis for the top120 potential hits (**A**) and the top 50 drug-like hits (**B**). The definition, numeric cutoff, and range for each risk score can be found in Methods 3.4.

**Figure 6 molecules-28-08034-f006:**
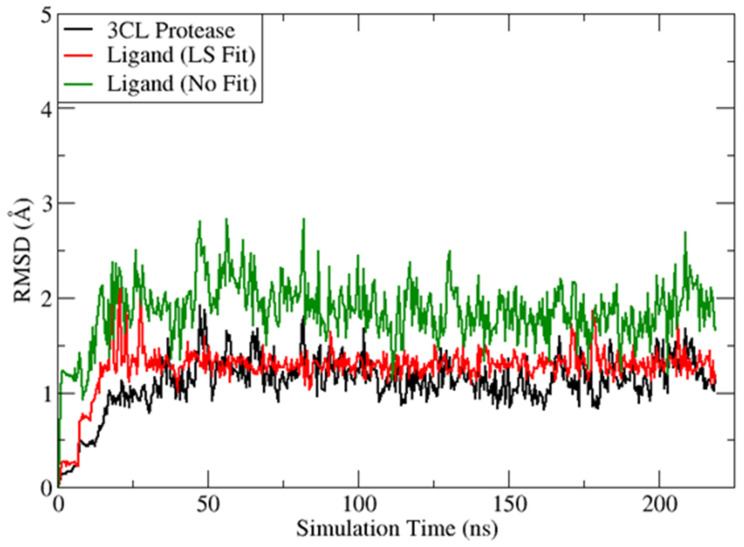
RMSD of the 3CL-pro protein and KB-2 (**PubChem CID: 14630497**) ligand with or without least square from the binding simulation for 218 ns. LS, least square fit. RMSD, Root Mean Square Deviation.

**Figure 7 molecules-28-08034-f007:**
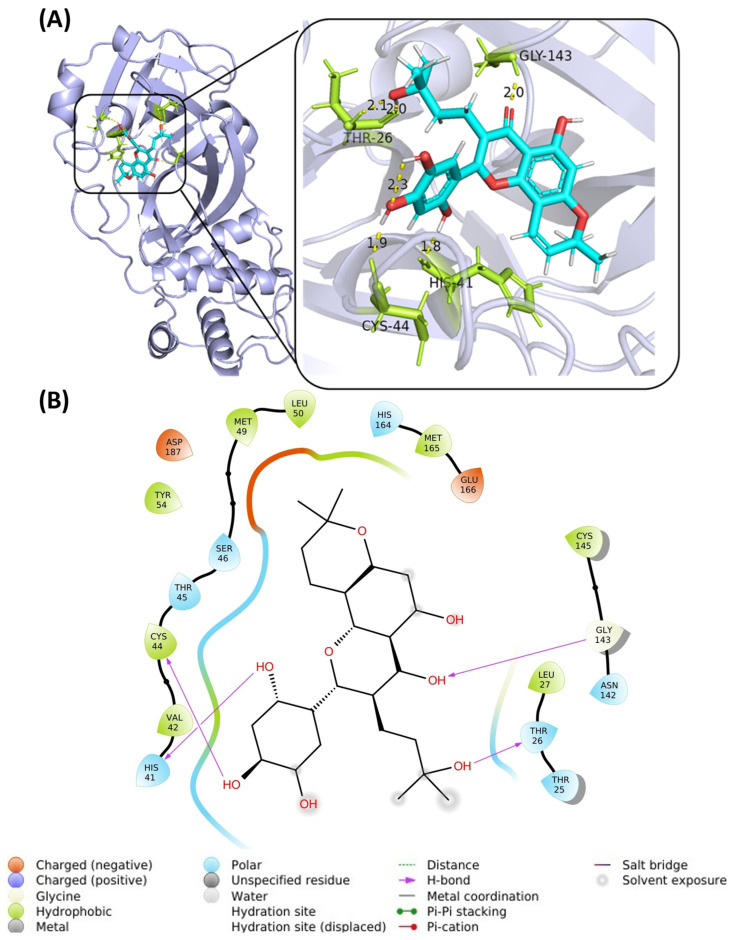
Docking pose (**A**) and 2D interaction patterns (**B**) of KB-2 (**PubChem CID: 14630497**) after molecular dynamics simulation in a complex with 3CL-protease (PDB: 6M2N).

**Figure 8 molecules-28-08034-f008:**
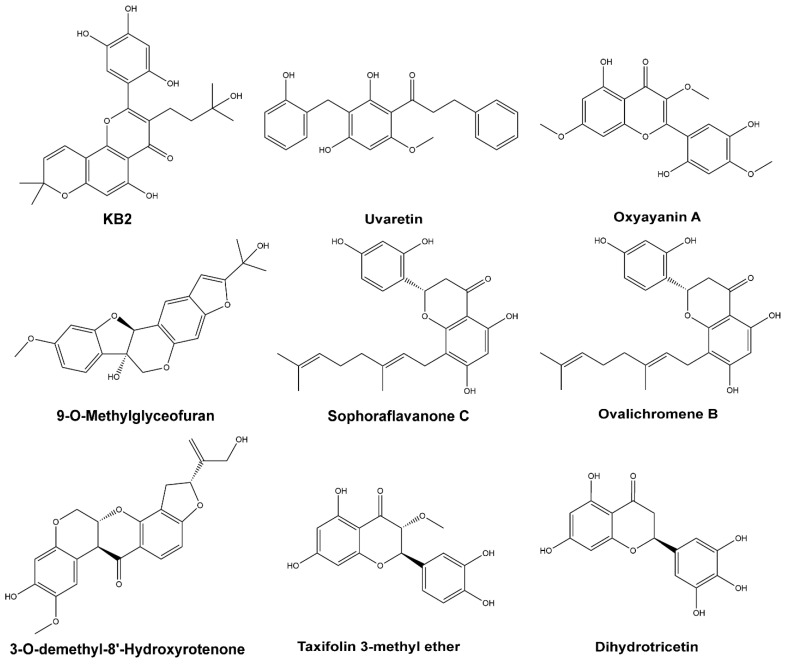
Molecular sturcture of the selected top nine flavonoids. The PubChem ID for each compound is listed in Table 2.

**Figure 9 molecules-28-08034-f009:**
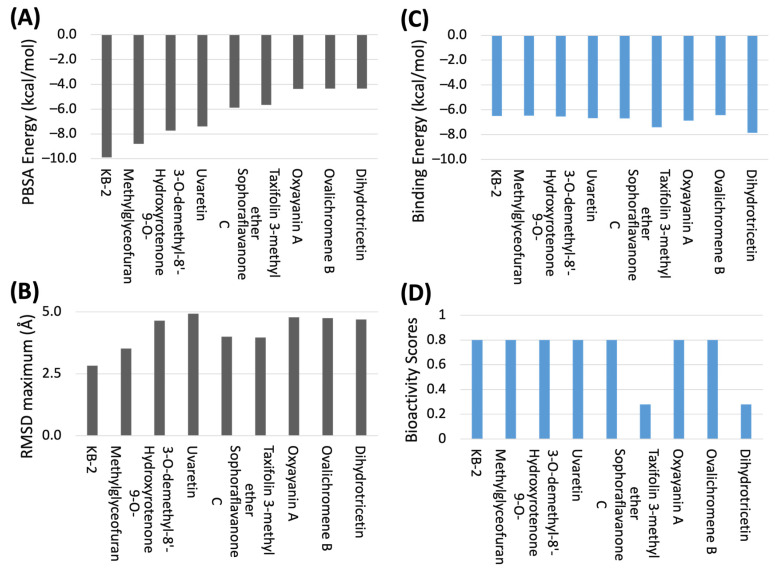
Predicted binding ability targeting at 3CL-pro for the selected top9 flavonoids obatained from molecular dynamics simulation (black, (**A**,**B**)) and from consensus-based Machine Lerarning models predictions (blue, (**C**,**D**)), which constituted the four important screening criteria of flavonoid hits. MM/PBSA energy, molecular mechanics energies combined with the Poisson–Boltzmann and surface area continuum solvation. RMSD, root-mean-square deviation.

**Table 1 molecules-28-08034-t001:** Dataset preparation and characteristics.

	Training Set A	Training Set B	Prediction Set
Sources	ChEMBL Database [38]	Library Composition [17]	Flavonoids Metabolites Database [39]
Usage	Regression Training	Classification Training	Prediction and Screening
Input	Ligand-residue energy interaction profiles (57 interacting residues)		
Bioactivity Values	Half Maximal Inhibitory Concentration, IC_50_ (nM)	Normalized Inhibition, inhibition %	
Output (Transformed)	Continuous Free Energy, ΔE_exp_ (kcal/mol)	Binary Bioactivity Label	
# Total Compounds	1240	8702	6961
# Processed Compounds	1118	7860	6001
# Qualified Compounds	1101	6059	
N_decoys_/N_actives_	1002/99	5729/330	
# Balanced Subsets	10 (A1–A10)	20 (B1–B20)	

Processed compounds refer to the retained compounds during the process of calculating energy interaction profiles. Qualified compounds refer to those yielding better docking scores on activation sites compared to on the dimerization sites. The symbol “#” refers to the number of compounds/datasets. N_decoys_/N_actives_ refers to the ratio of decoys to the actives in each training sets.

**Table 2 molecules-28-08034-t002:** List of the top nine compounds with ADMET risk scores and MM/PBSA binding energy (kcal/mol) as well as maximum value of No Fit RMSD (Å) for 3CL-pro calculated by MD simulation.

Compound Name	Structure Class	PubChem CID	PBSA Energy (kcal/mol)	RMSD Max (Å)	Full ADMET Scores
KB-2	Flavones	14630497	−9.89	2.83	3.500
9-*O*-Methylglyceofuran	Isoflavonoids	44257401	−8.81	3.52	3.016
3-*O*-demethyl-8’-Hydroxyrotenone	Isoflavonoids	44257401	−7.73	4.65	4.220
Uvaretin	Chalcones	73447	−7.41	4.93	4.010
Sophoraflavanone C	Flavanones	85403243	−5.89	4.01	2.691
Taxifolin 3-methyl ether	Dihydroflavonols	14794885	−5.66	3.97	4.163
Oxyayanin A	Flavonols	5281676	−4.38	4.79	2.500
Ovalichromene B	Flavanones	10981007	−4.35	4.75	5.503
Dihydrotricetin	Flavanones	5258991	−4.34	4.70	1.188

**Table 3 molecules-28-08034-t003:** ADMET and drug-likeness screening criteria and thresholds.

Risk Model	Thresholds (Range)	Criteria
Full ADMET Risk	7.0 (0–22.0)	Exceeds 7 for 10% of a focused WDI subset when ALL component risks are included.
Absorption Risk (Absn Risk)	4.0 (0–8.0)	Exceeds 4 for 9% of a focused WDI subset.
Lipinski’s Rule of 5 (Ro5)	1.0 (0–5.0)	Exceeds 1 for 8% of a focused WDI subset.
Risk connected with P450 oxidation (CYP Risk)	2.0 (0–6.0)	Exceeds 2.0 for 10% of a focused WDI subset.
Risk of mutagenicity (MUT Risk)	1.2 (0–5.4)	Exceeds 1.2 for 12% of a focused WDI subset.
Enhanced risk of mutagenicity (MUT_x)	1.0 (0–4.0)	Exceeds 1.0 for 12% of a focused WDI subset.
Risk connected with predicted toxicity (TOX Risk)	2.0 (0–6.0)	Exceeds 2.0 for 9% of a focused WDI subset.

Different risk models represent several different components assessed in the Full ADMET Risk model and within each model: the range of risk scores implies the number of risk rules that make up the corresponding risk criterion. In the case of the CYP Risk filter, for example, the model is comprised of six rules, including the CYPs 1A2, 2C9, 2C19, 2D6,3A4 and liver microsomal clearance, each with an associated weight of one. The score indicates the number of potential specific problems a compound might have. WDI, World Drug Index.

## Data Availability

All data used for the IP-SF development and prediction were provided as a part of Appendix A. The descriptions and processes of targets and compounds, feature representation, the model construction process, values of performance metrics for each model, ADMET risks analysis, and molecular dynamics analysis are presented in the Appendix A. Note that the identifiers of the compounds in each dataset corresponded to their data sources.

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
