# Peer review of "In Silico Screening of Natural Flavonoids against 3-Chymotrypsin-like Protease of SARS-CoV-2 Using Machine Learning and Molecular Modeling"

_molecules, 2023, doi:10.3390/molecules28248034_

Round 1
Reviewer 1 Report
Comments and Suggestions for Authors
In this manuscript, Cai et al. employ machine learning to screen natural flavonoids against the 3-Chymotrypsin-like Protease of SARS-CoV-2, aiming to discover new methods to combat SARS-CoV-2. They conduct the screening of 6001 flavonoids using ligand-residue energy interaction profiles (IPS). Eventually, they identify nine flavonoids that show potential effects against SARS-CoV-2. While their approach involves a large dataset and is operationally straightforward and rapid, I have reservations about the nine potential targets identified because the authors did not validate the actual effects of these flavonoids through biological experiments in the article.
The ultimate goal of such screenings is application, and the affinity analysis of drugs can only indicate potential therapeutic effects. Whether these effects truly exist needs confirmation through specific biological experiments. I recommend that the authors supplement their work with concrete experiments involving cells, mice, or clinical trials to demonstrate the efficacy of the nine identified targets before resubmitting the manuscript. Mere binding ability is insufficient.
Author Response
Dear Editors:
We sincerely thank the reviewers for careful reading of our manuscript and insightful suggestions on further improving the manuscript. We have addressed the minor issues, updated the references, and highlighted all revisions in the revised manuscript version.
We have carefully revised the manuscript properly addressed both reviewers’ comments. Below, we detail our responses to each comment and the corresponding revisions made to the manuscript. The comments from the reviewers are shown in blue color, our response in black, and the revised text in red.

Reviewer 2 Report
Comments and Suggestions for Authors
Please find attached.

Author Response

(The authors gave the same response as above.)
